# Assessment of the Accuracy in Measuring the Enamel Thickness of Maxillary Incisors with Optical Coherence Tomography

**DOI:** 10.3390/diagnostics12071634

**Published:** 2022-07-05

**Authors:** Hiroshi Miyagi, Kyosuke Oki, Yoshihiro Tsukiyama, Yasunori Ayukawa, Kiyoshi Koyano

**Affiliations:** 1Section of Implant and Rehabilitative Dentistry, Division of Oral Rehabilitation, Faculty of Dental Science, Kyushu University, Fukuoka 812-8582, Japan; hiroshi-miyagi@hotmail.co.jp (H.M.); ayukawa@dent.kyushu-u.ac.jp (Y.A.); 2Section of Fixed Prosthodontics, Division of Oral Rehabilitation, Faculty of Dental Science, Kyushu University, Fukuoka 812-8582, Japan; 3Section of Dental Education, Faculty of Dental Science, Kyushu University, Fukuoka 812-8582, Japan; 4Division of Advanced Dental Devices and Therapeutics, Faculty of Dental Science, Kyushu University, Fukuoka 812-8582, Japan; koyano@dent.kyushu-u.ac.jp

**Keywords:** optical coherence tomography, refractive index, enamel thickness

## Abstract

Although the clinical assessment of enamel thickness is important, hardly any tools exist for accurate measurements. The purpose of this study was to verify the precision of enamel thickness measurements using swept-source optical coherence tomography (SS-OCT). Human extracted maxillary central and lateral incisors were used as specimens. Twenty-eight sites were measured in each specimen. The optical path length (OPL) at each measurement site was measured on the OCT images, and enamel thickness (*e1*) was calculated by dividing OPL by the mean refractive index of enamel, 1.63. The specimens were then sectioned, and a light microscope was used to measure enamel thickness (*e2*). *e1* and *e2* were then compared. Measurement errors between *e1* and *e2* for the central and lateral incisors were 0.04 (0.02; 0.06) mm and 0.04 (0.02; 0.07) mm [median value: (25%, 75% percentile)], respectively. No significant differences between measurement sites were noted for measurement errors between *e1* and *e2*. These results demonstrate that OCT can be used for noninvasive, accurate measurements of enamel thickness.

## 1. Introduction

Restoration of porcelain laminate veneer is used to improve discolored teeth and malformed teeth, as it is a treatment that minimizes invasion of teeth and restores aesthetics. In the past, the elimination and chippage of porcelain laminate veneer have been reported as causes of prognosis and reintervention of treatment [1,2]. Since the thickness of laminate veneer is required to prevent both fracture and the transmission of unideal tooth surface color, dentists tend to overprepare the surface of the tooth to fabricate thick laminate veneer. This may cause the adhesion of laminate veneer to dentin, however, the different compositions of enamel and dentin, dentinal tubular structure and intratubular fluid movement in dentin [3,4] necessitate different surface treatments before the application of adhesive resin cements. That is, phosphoric acid etching for enamel surface and self-etching primer treatment for dentin are recommended. Even with selective surface pretreatment, however, studies comparing the adhesive strength of resin cements with enamel and dentin have indicated that stronger adhesive strength is achieved for enamel than dentin [5,6]. For this purpose, it is important to limit tooth preparation within enamel and to avoid laminate veneer adhering to dentin to prevent detachment. In fact, a systematic review paper indicated that veneer preparation in dentin was reported to affect survival adversely [7].

In addition to the reduction of the risk of detachment, porcelain veneers bonded to enamel offer stronger fracture resistance than those bonded to dentin [8]. Thus, from the viewpoint of both detachment and fracture, accurate measurement of enamel thickness before the tooth preparation can reduce the risk of overpreparation to dentin. In clinical dentistry, objective methods for measuring enamel thickness include computed tomography (CT) and ultrasonography [9,10,11]. Dental CT has a voxel size of approximately 0.1 mm generally and can therefore be used to measure enamel thickness with high precision. However, the use of CT to measure enamel thickness also poses various problems clinically, such as radiation exposure, time lag and artifacts caused by metal objects. Because ultrasonography has a lower resolution, it is difficult to ensure that enamel thickness is measured accurately with ultrasonography [10].

In recent years, optical coherence tomography (OCT) has been developed. OCT is a noninvasive imaging system that can utilize near-infrared light to produce high-resolution cross-sectional images of internal structures [12]. OCT is based on the concept of low-coherence interferometry [13]. In OCT, the laser source is projected onto the specimen and the backscattered signal intensity from inside the scattering medium is measured, with the scattering and reflection of light within the specimen displayed as imaging depth. Because imaging depth in OCT involves the calculation of optical attenuation to tissue (absorption, scattering), it is limited to approximately 2–3 mm in most tissue types [14,15]. However, OCT image resolution is 10–100 times more refined than ultrasonography. OCT also offers a great advantage over conventional microscopy, which requires that specimens be processed to undergo analysis [16,17]. Swept-source OCT (SS-OCT) in particular offers sensitivity, a high scan rate and an increased signal-to-noise ratio [18,19].

OCT was first reported by Fujimoto et al. in 1991 [13]. OCT has been used in many clinical applications, including gastroenterology [20], ophthalmology [21], dermatology [22] and dentistry [23]. In the field of dentistry, the first in vitro images of dental hard and soft tissues in a porcine model were reported in 1998 [24]. Later, the in vivo imaging of human dental tissue was shown [25]. In recent years, the usefulness of OCT is also being demonstrated for procedures such as the diagnosis of dental caries [26], the precision matching of resin-based restorations [27], and material testing [28]. OCT can be used to measure optical path length (OPL) [29]. Because OPL is the product of the refractive index and actual thickness, OCT can be used to accurately calculate the refractive index from the actual thickness [30]. Furthermore, OPL can be divided by the refractive index to obtain actual specimen thickness [31]. Meng et al. reported that the refractive index of human enamel was about 1.63 with OCT [32]. After that, it has been reported by Hariri et al. that the refractive index of enamel is stable, because sections with and without trabecular structures exhibited similar refractive indices of 1.63 ± 0.02 and 1.62 ± 0.02, respectively (mean: 1.63 ± 0.02) [33].

However, no reports have verified the accuracy of OCT for measuring enamel thickness. The evaluation of enamel thickness before treatment would make it possible to obtain information necessary for prosthetic design to ensure that the procedure stops at the enamel level.

The purpose of the present study was to investigate the accuracy of enamel thickness measurements obtained by OCT by comparing them with actual measurements in extracted maxillary central and lateral incisors.

## 2. Materials and Methods

### 2.1. Specimen Preparation

The extracted teeth used in this study were selected from a pool of extracted teeth that had been stored in a physiological saline solution at room temperature immediately after they were extracted for periodontal disease from April 2013 to April 2014. The inclusion criteria were maxillary central incisors and lateral incisors without morphological defects from the crown to the root. Exclusion criteria were teeth with defects of enamel, such as caries and restorations in the labial cement–enamel junction (CEJ), marked attrition at the incisal edge, dental prostheses, defects of enamel at ≥10 areas in the measurement sites mentioned below, and enamel hypoplasia. In total, 16 central incisors and 10 lateral incisors were examined.

First, periodontal ligament of the extracted teeth was removed with dental hand scaler, and acrylic resin (Fit Resin; Shofu, Kyoto, Japan) was used to arrange each tooth with the tooth axis perpendicular in a resin block with a diameter of 10 mm and a height approximately halfway to the tooth root (Figure A1a).

The distance from the incisal edge on the middle of the labial surface in the longitudinal axis of the tooth to CEJ was divided into eight equal parts. The cross section of 1/8 of the distance from the incisal edge was defined as *H1*, 1/4 the distance as *H2*, 1/2 the distance as *H3*, 3/4 the distance as *H4*, and 7/8 the distance as *H5*. The six measurement regions comprised the labial surface, mesiofacial surface, mesial surface, lingual surface, distal surface, and distofacial surface in the respective five cross sections from *H1* to *H5*. However, the mesial and distal surfaces of *H5* were excluded because they hardly contained any enamel. Thus, a total of 28 measurement sites were set (Figure A2). A φ1-mm round bur was used to mark sites on approximately 1 mm from both sides of each of the 28 measurement sites on the specimens (Figure A1b). The central points of each of these marked sites were used as the measurement sites.

### 2.2. SS-OCT Imaging

The SS-OCT system used in this study (Dental SS-OCT, Prototype 1; Panasonic Healthcare, Co., Ltd., Tokyo, Japan) employs Fourier–domain OCT technology. The light source is a wavelength-sweeping laser with a rate of 30 kHz over a span of >100 nm centered at 1330 nm. Two-dimensional images have horizontal and axial resolutions of 20 μm and 12 μm, respectively, in air and pixel size of 2000 × 1019 (9.0 μm × 3.5 μm). This system has a handheld intraoral probe with a complementary metal-oxide semiconductor (CMOS) camera used to visualize the surface being scanned in real time.

The handheld probe was set at a fixed distance from the specimen stage so that the scanning light beam was oriented approximately perpendicular to the stage. The distance between the probe and the target tooth surface depended on the thickness of the tooth. Each specimen was placed with the tooth axis parallel to the stage and with the measurement region for each of the five cross sections from *H1* through *H5* on the uppermost surface so that OCT images could be obtained (Figure A1c and Figure A3). Then, one examiner measured the distance from the tooth surface to the dentino-enamel junction (DEJ) perpendicularly to the tooth surface from the center of the areas marked on each side of the measurement sites with the measurement tool of the OCT system, manually. Beforehand, we confirmed that the laser beam generated by this device sufficiently penetrated enamel and reached DEJ in a pilot experiment. The examiner had trained to recognize DEJ on OCT images by comparing an OCT image and a light microscopy image with 10 premolars as a preliminary experiment. Sites that could not be measured because of caries or crown restorations were excluded.

### 2.3. OCT Image Analysis

Because OCT is based on low-coherence interferometry, it can be used to measure OPL [29]. The length rate changes of OCT images are required to convert optical depth values measured with OCT into actual depth values [30,31]. Figure A4 shows a schematic diagram of enamel on the specimen stage and an OCT image of change in the vertical position of the specimen stage brought about by the specimen. The vertical position of the specimen surface is *Z0*, the vertical position of the specimen stage is *Z1*, the specimen thickness is d, and the specimen stage position imaged through the tissue is *Z2*. Figure A4b shows that specimen thickness could be determined by subtracting the vertical position of the specimen stage without the specimen (*Z1*) from the specimen surface (*Z0*) with OCT images. In addition, OPL could be measured by subtracting the vertical position of the specimen stage imaged through the specimen (*Z2*) from the specimen surface position (*Z0*). Therefore, we can express the OPL of the specimen as *Z2*–*Z0* and the actual specimen thickness as *Z1*–*Z0*. Assuming the specimen length rate of change is *cs*, the following formula for the relationship between thickness and the length rate of change can be obtained.
(1)$$e=Z1-Z0=\frac{Z2-Z0}{cs}$$

The enamel length rate of change is the same as the refractive index of enamel and is relatively fixed at 1.63, regardless of enamel trabecular structure [33]. Therefore, enamel thickness can be expressed with the following formula.
(2)$$e=\frac{Z2-Z0}{1.63}$$

In the present study, this formula was used to calculate enamel thickness (*e1*) from OPL on OCT images obtained as shown in Figure A3b.

### 2.4. Light Microscopy Measurements

After OCT measurement, each specimen was completely covered in acrylic resin, and a diamond-saw microtome (Leica SP1600; Leica, Wetzlar, Germany) was used to prepare sections of *H1*–*H5*. This device had an annular innerhole saw, 8.3 cm in diameter and 300 μm in thickness. Next, a light microscope (Keyence BIOREVO BZ-9000; Keyence, Osaka, Japan) was used to measure enamel thickness by measuring the distance from the center of the markings on the left and right to DEJ (*e2*) perpendicular to the tangent of the middle of the marks (Figure A1d,e and Figure A5). *e2* was measured by the same examiner as the OCT measurement. Sites with caries or crown restorations were excluded from measurement.

### 2.5. Statistical Analysis

The above 28 measurement sites were set up to include thick and thin enamel and strongly and slightly curved enamel. Data were analyzed with the Kruskal–Wallis test to determine significant differences among measurement sites in differences between values calculated with mean refractive index (1.63) on OCT images (*e1*) and actual measurement values (*e2*) determined using a light microscope. The level of significance was set at α = 0.05. Statistical analysis was performed using IBM SPSS statistics 22.0 (Chicago, IL, USA).

## 3. Results

The number of valid measurement sites for 16 maxillary central incisors and 10 maxillary lateral incisors was 363 and 208, respectively. Table 1 and Table 2 shows the median of the difference between *e1* and *e2* values for maxillary central incisors and lateral incisors, respectively.

No significant differences in difference between *e1* and *e2* were observed for either central or lateral incisors (*p* = 0.141, 0.542, respectively) for each measurement site. Figure 1 shows the difference between *e1* and *e2* values of all measurement sites.

Figure 2 shows the average value and the standard deviation of *e2* and the number of valid measurement sites for each measurement region.

## 4. Discussion

In this study, SS-OCT appeared to be a potential tool that can measure enamel thickness accurately. Other objective methods for measuring enamel thickness include CT and ultrasonography. Ultrasonography has a lower resolution than OCT, and a past report that measured enamel thickness using A-scan reported that the measurement error with actual measurements was ≤10% [11]. In this study, we found that in the labial surface *H2* site of the central incisor, which was almost the same site as that used in a past report with an ultrasound system [11], measurement precision was higher with OCT (median value: 2.5%) than with the ultrasound system. 

It has been reported that tooth surface hydration conditions influence the signal intensities of DEJ on OCT images, and OCT images of teeth after 10 min of air blowing had greater signal intensity of DEJ than those that were wet or were immediately subsequent to air blowing [34]. In this study, OCT measurement was performed within 10 min after wiping the water on the tooth surface to ensure that tooth surface hydration condition was not much different from the clinical condition. Therefore, it seems that enamel thickness can be measured with SS-OCT with almost the same accuracy in this study.

The measurement error seen may have been caused by the enamel refractive index being calculated as a mean value of 1.63, which did not take into consideration differences in refractive indices for each tooth or measurement site. In particular, the use of a refractive index value of 1.63 for sites with thick enamel may have led to increased measurement error, but no significant differences were noted between measurement errors for each measurement site. Moreover, because the tooth surface is curved, the emitted laser light beam was diagonal to some measurement sites. In this study, the scanning light beam was perpendicular to the tooth surface to be measured but was not always perpendicular to each measurement site. This may have affected measurement errors. However, no significant differences in the extent of measurement errors were noted between sites near the incisal edge, where the laser light beam was oblique and directed from the labial surface center, to the tooth cervix region, where the scanning light was directed perpendicularly. This suggested that the use of the enamel refractive index of 1.63 was appropriate.

One of the possible clinical applications of OCT is the measurement of enamel thickness before the preparation of porcelain laminate veneer. When preparing porcelain laminate veneers, which requires enamel reduction of approximately 0.3–0.5 mm [35,36] and diamond depth-cut bur used in porcelain veneer preparation to have a diameter of 100 μm units generally, data of 10 μm units is not important for clinical judgments. Errors in the enamel refractive index and the scanning beam angle of incidence may affect OCT measurement errors. However, the fact that the median value of the measurement error was 0.04 mm in this study suggests that OCT can be used to measure enamel thickness with high precision. Thus, our results indicate that OCT could be clinically useful in the measurement of enamel thickness. OCT has some possibility for dentists to measure enamel thickness simply in real-time and to reduce enamel.

The limitation of the present study is that the measurement was done under room light. In case of clinical situation, the brightness and color temperature in the oral cavity may vary. In addition, laser beam application perpendicular to the tooth surface is not necessarily easy in clinical use, especially in the case of dental crowding. These have a possibility of influencing the measurement value. Next, in the present study we used tooth surfaces without caries. As well as caries, white spot and the decalcification of enamel under the surface layer are often observed in clinical situations. No information is provided in such cases in the present study. Further modification of the device and data accumulation using teeth with damaged surfaces are expected.

## 5. Conclusions

Within the limitations of this in vitro study, OCT has a possibility of providing the accurate measurement of enamel thickness in the maxillary central incisors and lateral incisors nondestructively. This study was performed under a controlled situation using teeth with intact surfaces. A modified method and auxiliary devices that can provide better measurement condition is needed to enhance reliability. In addition, the data should be collected using teeth with damaged surfaces.

## Figures and Tables

**Figure 1 diagnostics-12-01634-f001:**
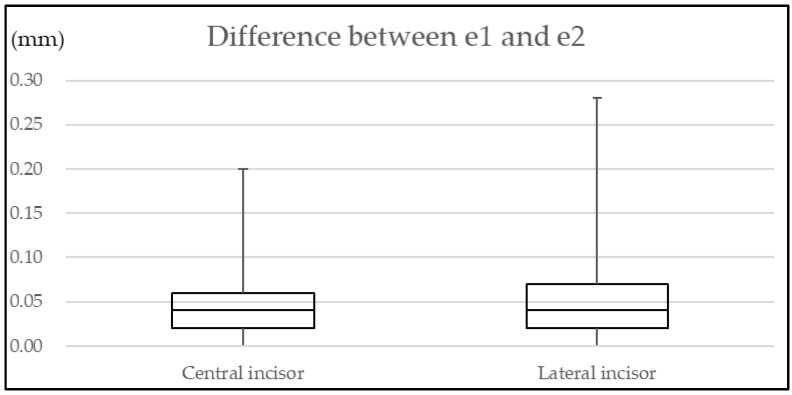
Difference between *e1* (measured by OCT) and *e2* (measured by microscopy).

**Figure 2 diagnostics-12-01634-f002:**
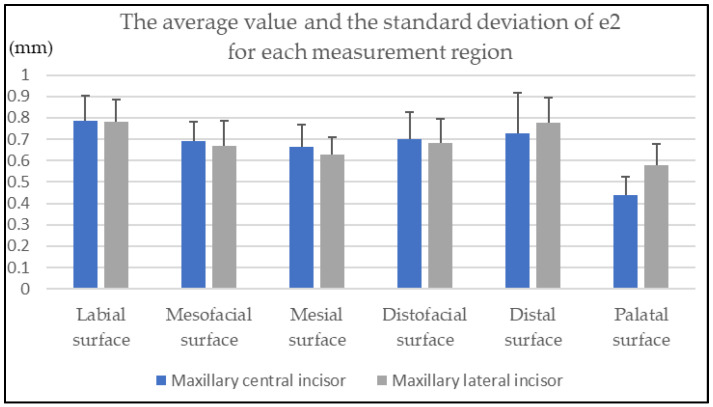
The average value and the standard deviation of *e2* (measured by microscopy) for each measurement region.

**Table 1 diagnostics-12-01634-t001:** Median of difference between *e1* (measured by OCT) and *e2* (measured by microscopy) for maxillary central incisors (mm).

	Labial Surface	Mesiofacial Surface	Mesial Surface	Distofacial Surface	Distal Surface	Palatal Surface
*H1*	0.02	0.05	0.03	0.05	0.06	0.03
*H2*	0.02	0.04	0.02	0.03	0.02	0.04
*H3*	0.04	0.05	0.05	0.07	0.05	0.03
*H4*	0.02	0.03	0.06	0.04	0.06	0.04
*H5*	0.04	0.03	-	0.04	-	0.04

**Table 2 diagnostics-12-01634-t002:** Median of difference between *e1* (measured by OCT) and *e2* (measured by microscopy) for maxillary lateral incisors (mm).

	Labial Surface	Mesiofacial Surface	Mesial Surface	Distofacial Surface	Distal Surface	Palatal Surface
*H1*	0.04	0.03	0.08	0.04	0.05	0.06
*H2*	0.07	0.05	0.07	0.07	0.04	0.06
*H3*	0.03	0.05	0.05	0.02	0.05	0.04
*H4*	0.05	0.04	0.05	0.03	0.04	0.05
*H5*	0.03	0.03	-	0.03	-	0.04

## Data Availability

The primary data that support the results described here are available from the corresponding author upon reasonable request.

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
