# Peer review of "Assessment of the Accuracy in Measuring the Enamel Thickness of Maxillary Incisors with Optical Coherence Tomography"

_diagnostics, 2022, doi:10.3390/diagnostics12071634_

Round 1

Reviewer 1 Report

Thanks for the article and I have the following comments:

1. "The handheld probe was set at a fixed distance from the specimen stage" so what is the fixed distance? This is also affecting the penetration depth, right? What is the penetration depth now? 

2. Why not using other method such as micro CT to find the enamel thickness? Would microtome cutting induce some errors in the sectioning? Perhaps you should mention the blade (e.g. size, thickness) that you have been used in the section so that we know how many tooth structures were lost during the sectioning. 

Author Response

Dear reviewer, thank you very much for spending your valuable time.

  1. "The handheld probe was set at a fixed distance from the specimen stage" so what is the fixed distance? This is also affecting the penetration depth, right? What is the penetration depth now?

-Probe was hold, not free hand, but fixed on a base that allowed light beam to strike the specimen perpendicularly. In this case fixed distance meant that the distance from prove to stage was constant (and the distance from prove to tooth varied, depended on the thickness of tooth).

We do not have detailed penetration depth but confirmed that the beam generated by this device sufficiently penetrated enamel and reached dentino-enamel junction. In a pilot experiment, we tested a number of tooth and confirmed it.

We modified Materials and Methods 2.2. SS-OCT imaging in the revised manuscript.

2. Why not using other method such as micro-CT to find the enamel thickness? Would microtome cutting induce some errors in the sectioning? Perhaps you should mention the blade (e.g. size, thickness) that you have been used in the section so that we know how many tooth structures were lost during the sectioning. 

-In the present study, we used a diamond-saw microtome. This device is exclusively used for cutting of metal, ceramic and non-decalcified hard tissue (bone and tooth). This has an annular innerhole (8.3 cm in diameter) saw in the thickness of 300 μm. The stage of specimen can move very slowly. Gentle cutting using extremely thin saw and slower specimen movement to saw can minimize the damage of specimen. We cut the tooth perpendicularly to tooth axis and confirmed that no structures at the surface of enamel were lost. As commented by the reviewer, microCT analysis to measure enamel thickness is informative, but we thought direct measurement is more accurate than the measurement using X-ray.

The information about the dimension of saw is added in the revised text.

Reviewer 2 Report

The article entitled “Assessment of the accuracy in measuring the enamel thickness of maxillary incisors with optical coherence tomography” aimed to verify the precision of enamel thickness 16 measurements using swept-source optical coherence tomography (SS-OCT).

The paper is in line with journal’s aim, moreover, Authors have well revised several issues; however, I ask authors to add aimed to provide an overview of available silver-treated dental implants and some key concepts.

-       In the title it is necessary to specify the type of paper

-       In the introduction section the authors introduced the main topic in relation to the positioning of veneers on the incisors, why not also discuss the optimization of adhesion materials and surface pre-treatments to reduce the thickness of the materials themselves and improve the fit of the veneers ? (please, see and discuss DOI

-       10.1002 / chem.201302704)

-       The results are confusing, please rewrite them and simplify the tables (too many numbers and non-linear formatting)

-       Conclusions cannot be reduced to a sentence: you must improve them highlighting the limits and the future insights pointed out from this article.

In the discussion it is necessary to better investigate the clinical limitations regarding the use of OCT in the evaluation of enamel thicknesses, please discuss the evidence present in this regard.

According to this Reviewer’s consideration, novelty and quality of the paper, publication of the present manuscript is recommended after minor revision.

Author Response

Dear reviewer, thank you very much for spending your valuable time.

 In the title it is necessary to specify the type of paper

-We add type of paper (Article) in the revised version.

In the introduction section the authors introduced the main topic in relation to the positioning of veneers on the incisors, why not also discuss the optimization of adhesion materials and surface pre-treatments to reduce the thickness of the materials themselves and improve the fit of the veneers ? (please, see and discuss DOI10.1002 / chem.201302704

-Thank you for your comment. We revised sentences regarding the importance of the optimization of adhesion materials and surface pre-treatments for preventing the detachment. The reviewer recommended to discuss DOI10.1002 / chem.201302704. This recommended paper was studied about the dispersion of carbon nanotubes in a surfactant hydrogel and we cannot correlate this paper with our research.

 The results are confusing, please rewrite them and simplify the tables (too many numbers and non-linear formatting)

-In the revised manuscript, percentile value in tables 1 & 2 are deleted and some tables are changed to graphs. All omitted data can be obtained by readers, as indicated in Data Availability Statement, as “The primary data that support the results described here are available from the corresponding author upon reasonable request.”

 Conclusions cannot be reduced to a sentence: you must improve them highlighting the limits and the future insights pointed out from this article.

-In the revised manuscript, we add sentences in conclusion section regarding the limitation and future insigiht of this study.

 In the discussion it is necessary to better investigate the clinical limitations regarding the use of OCT in the evaluation of enamel thicknesses, please discuss the evidence present in this regard.

-In the revised manuscript, we add a paragraph at the end of discussion regarding the limitation of this study.